# Clinical Characteristics and Prognostic Factors of Non-Infectious Cerebral Venous Sinus Thrombosis

**DOI:** 10.3390/jcm11206096

**Published:** 2022-10-16

**Authors:** Yu-Chieh Chen, Chun-Wei Chang, Hsiu-Chuan Wu, Chiung-Mei Chen, Chien-Hung Chang, Kuo-Hsuan Chang

**Affiliations:** Department of Neurology, Chang Gung University School of Medicine, Chang Gung Memorial Hospital, Linkou Branch, Taoyuan City 333, Taiwan

**Keywords:** cerebral venous thrombosis, stroke, cerebral infarction, intracranial hemorrhage, D-dimer, prognosis

## Abstract

Non-infectious cerebral venous thrombosis (CVT) is an uncommon type of cerebrovascular disease that usually affects young patients. It occurs frequently in female patients, probably due to the association of sex-specific risk factors for coagulopathies. Currently, the prognostic factors of CVT remain unclear. We retrospectively reviewed the clinical characteristics among 260 CVT patients, including 147 females and 113 males. A favorable clinical outcome was defined by the scores of the modified Rankin Scale (mRS) ≤ 2 at hospital discharge, while a poor clinical outcome was defined by an mRS score of 3 to 6. A headache (28.5%) was the most frequent presentation. The most commonly affected sinus was the transverse-sigmoid sinus (59.6%). Most of the cases (78.5%) were treated with anticoagulants. One hundred and fifty-seven patients (60.4%) were discharged with favorable clinical outcomes. Consciousness disturbance (odds ratio: 5.01, *p* < 0.001) was associated with a poor clinical outcome. Patients with poor clinical outcomes demonstrated higher D-dimer levels on admission (4137.76 ± 3317.07 vs. 2476.74 ± 2330.87 ng/mL FEU, *p* = 0.029) and longer hospitalization days (31.81 ± 26.29 vs. 13.96 ± 8.82 days, *p* < 0.001) compared with favorable clinical outcomes. These findings provide important information of clinical characteristics and prognosis for CVT. Aggressive monitoring and treatment should be considered in CVT patients with poor prognostic factors.

## 1. Introduction

Non-infectious cerebral venous thrombosis (CVT) is an uncommon type of cerebrovascular disease that usually affects young patients, and approximately accounts for less than 1% of all forms of stroke in the adult population [1]. It occurs frequently in female patients, probably due to the association of sex-specific risk factors for coagulopathies including pregnancy and oral contraceptive use [2]. The diagnosis of CVT remains challenging due to the diversity of clinical presentations. Common clinical manifestations are headache, focal neurological deficits, seizure, and altered mental status, all of which could develop over several days [3]. Hypercoagulable status, such as heritable pro-thrombotic conditions, autoimmune diseases, oral contraceptive use, hormonal replacement therapy, pregnancy or puerperium, malignancy, and infections or chronic inflammatory disorders [3,4,5] are risk factors for CVT. Traumatic head injury and arteriovenous malformations are also associated with the development of CVT [6,7]. CVT is occasionally seen in patients receiving COVID-19 vaccinations [8,9]. Patients with CVT may have multiple predisposing factors [3], thus a thorough investigation is needed.

Neuroimaging is mandatory for diagnosing CVT. Non-contrast computed tomography could be informative, showing specific imaging markers such as a dense triangle sign or cord sign, and ischemic territories that are not compatible with arterial territories or even with hemorrhagic transformation [7]. Computed tomography venography (CTV) and magnetic resonance venography (MRV) are confirmatory tools to reveal the location of the thrombosed sinus [10,11]. Anticoagulation with low-molecular-weight heparin (LMWH) is the mainstay therapy for CVT [11]. Treatments of the systemic diseases involving hypercoagulable states are also important. An endovascular thrombectomy is also considered in patients refractory to medical treatments [12]. Currently, the prognostic factors of CVT remain unclear. Many patients have favorable outcomes if treated properly in time; however, a few patients may have chronic sequelae or physical disabilities [13,14]. In this study, we demonstrated the clinical characteristics among CVT patients in a large cohort and explored the prognostic factors in patients with CVT. The results provide important information to clarify the pattern and outcome in patients with CVT.

## 2. Materials and Methods

### 2.1. Study Design and Participants

Patients with CVT were recruited from Chang Gung Research Database, which consists of branches of Chan Gung Memorial Hospital, including two tertiary referral centers and two secondary hospitals, from January 2001 to January 2021. The accuracy of the diagnoses of cerebral venous thrombosis of the subjects was confirmed by the codes of International Classification of Diseases (ICD)-9-CM (325.0, 437.6, 671.5) or ICD-10-CM (I67.6, O87.3) combined with the records of discharge summary. Diagnosis of CVT was confirmed with CT venography, MRI with magnetic resonance venography, or angiography. Comorbidity, vascular risk factors, and clinical course were identified after an in-depth review of the medical records by two board-certified neurologists (Chen YC and Chang KH). Laboratory tests including complete blood cell count, blood biochemistry, coagulation, and brain imaging studies were recorded. Functional outcome was defined by the modified Rankin Scale (mRS) and a dichotomous analysis was performed with a favorable or poor outcome. A favorable clinical outcome was defined by a satisfactory recovery and resumption of a normal functional life by the scores of mRS ≤ 2 at hospital discharge [15]. Poor clinical outcomes were defined by an mRS score of 3 to 6 [15]. We excluded patients diagnosed with cavernous sinus syndrome or thrombosis, patients diagnosed with acute lymphocytic leukemia (ALL) before the diagnosis of sinus thrombosis, vaccine-induced thrombotic thrombocytopenia (VITT), and those without image evidence of cerebral venous thrombosis. Although infection is a risk factor for CVT, it accounts for a smaller population when compared to non-infectious CVT, and required different treatment [16]. Therefore, we excluded patients with infectious CVT.

### 2.2. Statistical Analysis

Continuous variables with normal distribution were expressed as mean ± standard deviation, analyzed by a general linear model. Categorical variables were expressed as a number (percentage). Categorical variables were analyzed using the Chi-square test to identify risk factors by calculating OR, while continuous variables were analyzed by the Student‘s *t*-test. The odds ratio (OR) was defined as the ratio of the odds of an event in the group of poor clinical outcomes to the odds of it in the group of favorable clinical outcomes. A univariate logistic regression model was used to measure the odds ratio for favorable or poor outcome. All variables with *p* < 0.1 in the univariate logistic regression analysis entered multivariate logistic regression analysis to identify the risk factors for the favorable or poor outcome at discharge, and the variables with *p* < 0.05 were required for statistical significance. All statistical analyses were performed with SPSS statistics version 25.0 (IBM, Armonk, NY, USA), and the “R” statistical language system (version 4.0.3, R Fundation, Jaunpur, India)

## 3. Results

### 3.1. Demographic Characteristics of Patients with Cerebral Venous Thrombosis

We identified 260 patients with CVT, including 147 females and 113 males, and aged between 13 years and 91 years old. Hypertension (23.8%) was the leading comorbidity for CVT, followed by smoking (19.2%), oral contraceptives (12.3%), and diabetes mellitus (9.6%) (Table 1).

### 3.2. Clinical Presentations of Patients with Cerebral Venous Thrombosis

A headache (48.5%) was the most frequent clinical presentation in CVT patients, followed by limb weakness/numbness (26.2%), seizures (20.4%) and consciousness disturbance (20.4%, Table 2). The most common affected sinuses were transverse-sigmoid sinuses (66.5%, Figure 1a), followed by superior sagittal sinus (59.6%, Figure 1b), and cortical veins (14.2%). However, involvement of multiple sinuses was seen in 141 patients (54.2%). A headache and transverse-sigmoid sinus involvement were associated with favorable outcomes. Most of the patients (78.5%) were treated with anticoagulants and 77.1% of patients have a favorable outcome. Intracranial hemorrhage (46.5%) was the most common complication, followed by cerebral infarction (22.7%).

### 3.3. Identification of Factors Associated with Clinical Outcome

One hundred and three patients (39.6%) demonstrated a poor clinical outcome at discharge (mRS score ≥ 3), while 5% of patients died. Patients with a poor clinical outcome demonstrated higher D-dimer level on admission (poor vs. favorable: 4137.76 ± 3317.07 vs. 2476.74 ± 2330.87 ng/mL fibrinogen equivalent unit, *p* = 0.029) and longer hospitalization days (poor vs. favorable: 31.81 ± 26.29 vs. 13.96 ± 8.82 days, *p* < 0.001) compared with a favorable clinical outcome after adjusting for the identified risk factors (Table 3).

We use logistic regression analysis to identify the prognostic factors for poor clinical outcome in our study. In the univariate logistic regression model, hypertension (OR: 1.91, 95% confidence interval (CI): 1.07–3.41, *p* = 0.028), consciousness disturbance (OR: 7.99, 95% CI: 4.04–16.90, *p* < 0.001), intracranial hemorrhage (OR: 2.35, 95% CI: 1.42–3.92, *p* < 0.001), and cerebral infarction (OR: 2.17, 95% CI: 1.21–3.93, *p* = 0.010) were associated with poor clinical outcomes. On the other hand, a headache (OR: 0.21, 95% CI: 0.12–0.36, *p* < 0.001) and transverse-sigmoid sinus involvement (OR: 0.47, 95% CI: 0.28–0.80, *p* = 0.005) were associated with favorable clinical outcomes. After multivariate regression analysis, factors including consciousness disturbance (OR: 5.01, 95% CI: 2.07–12.80, *p* < 0.001), D-dimer level (*p* = 0.029), and length of hospital stay (*p* < 0.001) remained significantly higher in patients with a poor clinical outcome in comparison with those with a favorable clinical outcome (Table 4).

## 4. Discussion

By analyzing the clinical database of patients with CVT, we find CVT is most frequently developed in the transverse-sigmoid sinus. Hypertension, consciousness disturbance, intracranial hemorrhage, and cerebral infarction are associated with poor clinical outcomes, while a headache and transverse-sigmoid sinus involvement are associated with favorable clinical outcomes. Up to now, this is the largest retrospective cohort in Asian populations. Patients with poor clinical outcomes demonstrate higher D-dimer levels and longer hospitalization days compared to favorable clinical outcomes. Aggressive monitoring and treatment should be applied to the patients with the identified poor prognostic factors.

Consistent with other studies [5,13,17,18,19,20], our results show that a headache is the most common presentation of CVT [18]. The characteristics of a headache are mostly subacute onset and diffuse pain and may be aggravated by the Valsalva maneuver, but the patterns are mostly variable. Studies have shown that CVT patients can also present with isolated headaches without evident signs related to increased intracranial hypertension [21]. We also find consciousness disturbance as a factor for poor clinical outcomes. It has been shown that the involvement of the deep cerebral venous system with extended venous thrombosis often leads to consciousness disturbance, and the subsequent poor clinical outcome [13]. However, thrombosis in the deep cerebral venous system is not associated with a poor clinical outcome in other studies [22,23]. Future large cohort studies will be needed to confirm the clinical outcome of patients with thrombosis in the deep cerebral venous system.

In our study, around a half of patients have multiple sinuses involvement, and transverse-sigmoid and superior sagittal sinuses are the most common involved sinuses. The most common location of CVT in Caucasians is the superior sagittal sinus, followed by transverse sinuses [13]. The frequently affected sinus in the Portuguese is the transverse sinus, followed by sigmoid sinuses and superior sagittal sinus [5]. In a recent French study [19], the most frequently occluded sinuses were the transverse sinus, followed by superior sagittal sinus. Transverse, superior sagittal, and sigmoid sinuses are the most commonly involved sinus of CVT in Chinese and Italian populations [17,20].

Our results showed D-dimer levels are elevated in CVT patients with poor clinical outcomes. D-dimer is a parameter commonly used for detecting deep vein thrombosis or pulmonary embolisms [24,25,26,27]. A previous study showed high sensitivity (97.8%) of D-dimer for diagnosing CVT [27,28]. D-dimer is the degradation product of the fibrin, and elevated D-dimer levels can be found in infection, pregnancy, malignancy, or thrombolytic treatment [29]. In CVT, a high level of D-dimer is probably due to the activation of the fibrinolytic mechanism, thus reflecting the extension of thrombosed sinuses. Whether D-dimer could be a prognostic marker for CVT needs to be validated in a large cohort.

Older age, and the development of cerebral infarctions and hemorrhage are associated with poor clinical outcomes in Turkish populations [5]. In Portuguese and Dutch populations, older age and cerebral hemorrhage are predictors of death or dependence in cerebral venous thrombosis patients [13]. Older age and venous infarction are associated with poor clinical outcomes (defined by mRS ≥ 2) in Italians [20]. However, the associations between poor clinical outcome and cerebral infarction or hemorrhage are not seen in our study. We do not recapitulate the difference in age between CVT patients with poor and favorable clinical outcomes.

There are several limitations in this study. There may be an unidentified selection bias toward patients with greater levels of disease severity in referral medical centers and loss of asymptomatic patients or errors of ICD coding. Patients who have mild symptoms and spontaneous resolution may not be identified as well. Patients with VITT caused by COVID-19 vaccines are not enrolled [8,9]. Future research to follow up the long-standing outcome in a large cohort is warranted.

## 5. Conclusions

Hypertension, consciousness disturbance, intracranial hemorrhage, cerebral infarction, high levels of D-dimer, and longer hospitalization days are poor prognostic factors for CVT. Headaches and transverse-sigmoid sinus involvement are associated with favorable clinical outcomes. Aggressive monitoring and treatment of CVT patients with poor prognostic factors should be considered.

## Figures and Tables

**Figure 1 jcm-11-06096-f001:**
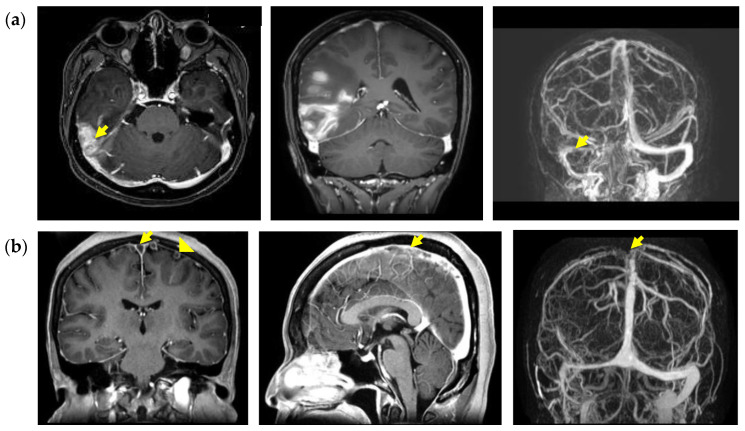
(**a**) A 27-year-old woman presented with a progressive headache for 2 days followed by vomiting. Brain MRA+ V showed right transverse-sigmoid sinus thrombosis with right temporal intracranial hemorrhage (upper row). (**b**) A 50-year-old woman presented with a progressive headache and cognitive dysfunction for 3 days. Brain MRA+ V showed superior sagittal sinus (arrow) and frontal cortical vein (arrowhead) thrombosis (lower row).

**Table 1 jcm-11-06096-t001:** Demographic characteristics of patients with cerebral venous thrombosis.

	Poor Clinical Outcome (*n* = 103)	Favorable Clinical Outcome (*n* = 157)	Total (*n* = 260)
**Female (%)**	60 (58.3%)	87 (55.4%)	147 (56.5%)
**BMI (kg/m^2^)**	24.23 ± 4.22	25.53 ± 4.85	25.04 ± 4.65
**Age (years)**	52.13 ± 19.38	45.39 ± 14.10	48.06 ± 16.69
**Hemoglobin (g/dL)**	12.51 ± 2.78	13.27 ± 2.54	12.96 ± 2.66
**Platelet Count (1000/uL)**	231.75 ± 105.23	248.94 ± 91.75	3149.06 ± 2878.81
**D-dimer (ng/mL FEU)**	4137.76 ± 3317.07	2476.74 ± 2330.87	242.04 ± 97.55
**Hospitalization days**	31.81 ± 26.29	13.96 ± 8.82	21.03 ± 19.89
**Hypertension**	32 (31.1%)	30 (19.1%)	62 (23.8%)
**Smoking**	18 (17.5%)	32 (20.4%)	50 (19.2%)
**Oral contraceptives**	8 (7.8%)	24 (15.3%)	32 (12.3%)
**Diabetes mellitus**	14 (13.6%)	11 (7.0%)	25 (9.6%)
**Pregnancy or postpartum**	9 (8.7%)	13 (8.3%)	22 (8.5%)
**Malignancy**	6 (5.8%)	4 (2.5%)	10 (3.8%)
**Hereditary coagulopathy**	3 (2.9%)	4 (2.5%)	7 (2.7%)
**Previous DVT or CVT**	2 (1.9%)	5 (3.2%)	7 (2.7%)
**Old ICH history**	1 (1.0%)	2 (1.3%)	3 (1.2%)
**PAOD**	2 (1.9%)	0 (0.0%)	2 (0.8%)
**Anemia**	1 (1.0%)	1 (0.6%)	2 (0.8%)
**Hyperthyroid**	1 (1.0%)	1 (0.6%)	2 (0.8%)
**Atrial fibrillation**	0 (0.0%)	1 (0.6%)	1 (0.4%)
**Thalassemia**	1 (1.0%)	0 (0.0%)	1 (0.4%)
**Adrenal insufficiency**	1 (1.0%)	0 (0.0%)	1 (0.4%)

BMI = body mass index; CVT = cerebral venous thrombosis, DVT = deep vein thrombosis, FEU = fibrinogen equivalent units; PAOD = peripheral artery occlusion disease, ICH = intracerebral hemorrhage.

**Table 2 jcm-11-06096-t002:** Clinical presentations of patients with cerebral venous thrombosis.

	**Poor Clinical** **Outcome (*n* = 103)**	**Favorable Clinical** **Outcome (*n* = 157)**	**Total (*n* = 260)**
**Signs and symptoms, *n* (%)**			
** Headache**	27 (26.2%)	99 (63.1%)	126 (48.5%)
** Limb weakness**	33 (32.0%)	35 (22.3%)	68 (26.2%)
** Seizure**	27 (26.2%)	26 (16.6%)	53 (20.4%)
** Consciousness disturbance**	41 (39.8%)	12 (7.6%)	53 (20.4%)
** Speech disturbance**	7 (6.8%)	6 (3.8%)	13 (5.0%)
** Visual disturbance**	5 (4.9%)	12 (7.6%)	17 (6.5%)
** Tinnitus**	0 (0.0%)	3 (1.9%)	3 (1.2%)
**Sinus thrombosis site, *n* (%)**			
** Transverse-sigmoid sinus**	58 (56.3%)	115 (73.2%)	173 (66.5%)
** Superior sagittal sinus**	66 (64.1%)	89 (56.7%)	155 (59.6%)
** Cortical vein**	12 (11.7%)	25 (15.9%)	37 (14.2%)
** Straight sinus**	16 (15.5%)	14 (8.9%)	30 (11.5%)
** Others**	14 (13.9%)	15 (9.6%)	29 (11.2%)
** Multiple sites**	56 (54.4%)	85 (54.1%)	141 (54.2%)
**Treatment, *n* (%)**			
** Anticoagulant**	83 (80.6%)	121 (77.1%)	204 (78.5%)
** Antiplatelet**	5 (4.9%)	11 (7.0%)	16 (6.2%)
** None**	15 (14.6%)	25 (15.9%)	40 (15.4%)
**Complications, *n* (%)**			
** Intracranial hemorrhage**	61 (59.2%)	60 (38.2%)	121 (46.5%)
** Cerebral infarction**	32 (31.1%)	27 (17.2%)	59 (22.7%)
** Arteriovenous fistula**	10 (9.7%)	11 (7.0%)	21 (8.1%)

**Table 3 jcm-11-06096-t003:** Differences between poor or favorable clinical outcome.

	**Poor Clinical** **Outcome (*n* = 103)**	**Favorable Clinical** **Outcome (*n* = 157)**	**T Value**	** *p* ** **Value**	**Adjusted *p* Value #**	**Effect** **Size** **†**
**BMI (kg/m^2^)**	24.23 ± 4.22	25.53 ± 4.85	1.761	0.085	0.50	
**Age (years)**	52.13 ± 19.38	45.39 ± 14.10	−3.241	0.002 *	0.70	
**Hemoglobin (g/dL)**	12.51 ± 2.78	13.27 ± 2.54	2.266	0.024 *	0.50	
**Platelet count (1000/uL)**	231.75 ± 105.23	248.94 ± 91.75	1.380	0.20	0.60	
**D-dimer (ng/mL FEU)**	4137.76 ± 3317.07	2476.74 ± 2330.87	−3.303	0.003 *	0.029 *	−0.419
**Hospitalization days**	31.81 ± 26.29	13.96 ± 8.82	−7.863	<0.001 *	<0.001 *	−0.997

BMI = body mass index; FEU = fibrinogen equivalent units; # *p* value adjusted by hypertension, headache, consciousness disturbance, transverse-sigmoid sinus, intracranial hemorrhage, and cerebral infarction by general linear model. * Statistically significant by comparison between poor and favorable clinical outcomes (*p* < 0.05). † Cohen’s *d* was calculated to estimate effect size.

**Table 4 jcm-11-06096-t004:** Analysis of the factors related to a poor clinical outcome.

	**Poor Clinical** **Outcome** **(*n* = 103)**	**Favorable Clinical Outcome** **(*n* = 157)**	**Univariate** **OR (95% CI)** ***p* Value**	**Multivariate** **OR (95% CI)** ***p* Value**
**Risk factor**				
** Oral contraceptives**	8 (7.8%)	24 (15.3%)	0.47 (0.19–1.04) 0.076	0.65 (0.20–1.96) 0.500
** Pregnancy or postpartum**	9 (8.7%)	13 (8.3%)	1.06 (0.42–2.56) 0.90	
** Hereditary coagulopathy**	3 (2.9%)	4 (2.5%)	1.15 (0.22–5.31) 0.90	
** Malignancy**	6 (5.8%)	4 (2.5%)	2.37 (0.66–9.45) 0.20	
** Previous DVT/CVT**	2 (1.9%)	5 (3.2%)	0.60 (0.09–2.85) 0.50	
** Hypertension**	32 (31.1%)	30 (19.1%)	1.91 (1.07–3.41) 0.028 *	1.22 (0.45–3.24) 0.700
** Diabetes**	14 (13.6%)	11 (7.0%)	2.09 (0.91–4.90) 0.083	1.00 (0.27–3.68) >0.900
** Smoking**	18 (17.5%)	32 (20.4%)	0.83 (0.43–1.55) 0.60	
** Anticoagulant**	83 (80.6%)	121 (77.1%)	1.23 (0.67–2.31) 0.50	
** Male**	43 (41.7%)	70 (44.6%)	0.89 (0.54–1.47) 0.70	
**Symptoms and signs**				
** Headache**	27 (26.2%)	99 (63.1%)	0.21 (0.12–0.36) <0.001 ***	0.50 (0.23–1.11) 0.088
** Limb numbness/weakness**	33 (32.0%)	35 (22.3%)	1.64 (0.94–2.88) 0.082	1.24 (0.53–2.91) 0.600
** Seizure**	27 (26.2%)	26 (16.6%)	1.79 (0.97–3.30) 0.061	0.94 (0.37–2.39) >0.900
** Consciousness disturbance**	41 (39.8%)	12 (7.6%)	7.99 (4.04–16.90) <0.001 *	5.01 (2.07–12.80) <0.001 *
** Speech disturbance**	7 (6.8%)	6 (3.8%)	1.84 (0.59–5.86) 0.30	
** Visual disturbance**	5 (4.9%)	12 (7.6%)	0.62 (0.19–1.72) 0.40	
** Tinnitus**	0 (0.0%)	3 (1.9%)	-	
**Thrombosis site**				
** Sagittal sinus**	66 (64.1%)	89 (56.7%)	1.36 (0.82–2.28) 0.20	
** Transverse-sigmoid sinus**	58 (56.3%)	115 (73.2%)	0.47 (0.28–0.80) 0.005 ***	0.57 (0.25–1.29) 0.200
** Cortical vein**	12 (11.7%)	25 (15.9%)	0.70 (0.32–1.43) 0.30	
** Straight sinus**	16 (15.5%)	14 (8.9%)	1.88 (0.87–4.09) 0.11	
** Others**	14 (13.9%)	15 (9.6%)	1.49 (0.68–3.25) 0.30	
** Multiple sites**	56 (54.4%)	85 (54.1%)	1.01 (0.61–1.66) >0.90	
**Complication**				
** Intracranial hemorrhage**	61 (59.2%)	60 (38.2%)	2.35 (1.42–3.92) <0.001 *	0.86 (0.29–2.53) 0.800
** Cerebral infarction**	32 (31.1%)	27 (17.2%)	2.17 (1.21–3.93) 0.010 *	1.02 (0.36–2.94) >0.900
** Arteriovenous fistula**	10 (9.7%)	11 (7.0%)	1.43 (0.57–3.52) 0.40	

CI = confidence interval, CVT = cerebral venous thrombosis, DVT = deep vein thrombosis, mRS = modified Rankin Scale. OR = odds ratio between poor and favorable clinical outcomes. * Statistically significant by comparison between poor and favorable clinical outcomes (*p* < 0.05). *** Statistically significant as favorable clinical outcome (*p* < 0.05).

## Data Availability

The datasets generated during the current study are available from the corresponding author on reasonable request.

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
