# Peer review of "Clinical Characteristics and Prognostic Factors of Non-Infectious Cerebral Venous Sinus Thrombosis"

_jcm, 2022, doi:10.3390/jcm11206096_

Round 1
Reviewer 1 Report
In the presented manuscript the authors analyze the outcome of CVT. This topic is actual and knowledges about risk factors of outcomes could be important in daily practice. The study is retrospective and some inclusion bias are presented.
There are some major concerns, that noted below:
1. It is not clear why in Table 1 patients are divided to 2 groups according to sex. It would be logically to compare demographic characteristics between good and poor outcome groups. Please clarify this point or change the table.
2. The same comment is applied to Table 2.
3. The Figure 1 does not correspond the sentence about frequency of involved sinuses. The Figure 1 show only MRI radiologic findings, but does not show rate of involved sinuses. Please remove or change this Figure.
4. In the figure 2 mRS score in noted on admission. Does the score mean the functional status on admission or before illness? The mRS score should not be used to describe the neurological deficit on admission. Please clarify. Additional comment to Figure 2 is the same as for Table 1 and Table 2 (presentation of results in male and female groups separately).
5. mRS score was estimated on discharge, while the length of hospital stay could be different. I would recommend to provide the mRS score on fixed day (for example day7) additionally.
6. Table 4 is unclear. I suggest to move the data from Table to Table 3 and performed regression analysis.
7. To identify the independent risk factors of poor outcome the multiregression analysis should be done. Please do this analysis.
8. After revision of results section the discussion section should be adopted according to results.
Minor concern: The sentence "We identified 260 patients with CVT, aged between 13-years-old 74 and 91-years-old" (line 74) repeated in Results section (lines 94 - 95), I would recommend to remove it from Methods Section.
Author Response
Response to reviewers
Reviewer 1
Comment 1: It is not clear why in Table 1 patients are divided to 2 groups according to sex. It would be logically to compare demographic characteristics between good and poor outcome groups. Please clarify this point or change the table. The same comment is applied to Table 2.
Response: Thank you for this suggestion. We summarized demographic data in table 1, and identified sex-specific factors for CVT. The demographic characteristics between good and poor outcome groups were demonstrated in Table 3 and 4.
Page 8, Line 190-194: While oral contraceptive use and pregnancy were well-known risk factors that are specific to female patients, risk factors for CVT in male populations remain inconclusive. By comparing the demographic features, we found that smoking was prevalent in male patients, and can be a considerable risk factors for CVT in male population.
Comment 2: The Figure 1 does not correspond the sentence about frequency of involved sinuses. The Figure 1 show only MRI radiologic findings, but does not show rate of involved sinuses. Please remove or change this Figure.
Response: We change Figure 1a and Figure 1b according to the sentence describing the affected sinuses.
Page 3, line 114-116: The most common affected sinuses were transverse-sigmoid sinus (66.5%, Figure 1a), followed by superior sagittal sinus (59.6%, Figure 1b) and cortical veins (14.2%).)
Comment 3: In the figure 2 mRS score in noted on admission. Does the score mean the functional status on admission or before illness? Please clarify.
Response: We recorded the mRS at day 7 and 30.
Page 6, Figure 2: Distribution of modified Rankin Scale (mRS) at day 7 and day 30 after onset of disease.
Comment 4: Table 4 is unclear.
Response: In Table 4, we performed general linear model (multiple regression) to adjust the categorical variables with different distribution between groups.
Page 8, Table 4: P value adjusted by hypertension, headache, conscious disturbance, transverse-sigmoid sinus, intracranial hemorrhage and cerebral infarction by general linear model.
Comment 5: To identify the independent risk factors of poor outcome the multiregression analysis should be done. Please do this analysis.
Response: We have done the univariate and multivariate logistic regression to identify the independent risk factors of poor outcome.
Page 6-7, Table 3
Comment 6: After revision of results section the discussion section should be adopted according to results.
Response: We revised the discussion section according to the results by multivariate regression.
Page 8-9, Line 211-240
Comment 8: The sentence "We identified 260 patients with CVT, aged between 13-years-old 74 and 91-years-old" (line 74) repeated in Results section (lines 94 - 95), I would recommend to remove it from Methods Section.
Response: Thank you for this suggestion. This sentence was removed.

Reviewer 2 Report
Table 1, 2 do not include the results of statistical tests.
The results of the statistical tests are not recorded according to scientific standards, e.g. t(24) = 3.94; p = 0.02. This applies to all tests used (chi-square, etc.).
The effect size for the statistical tests used has not been calculated (D Cohen, Phi, etc.). Each statistical test is marked by a specific effect size. The p-value alone is definitely not enough.
Univariate and multivariate logistic regression should be performed in the analysis. The results of this analysis should be included in a summary table. Based on this, more reliable conclusions can be drawn.
In the introduction, the authors did not include available articles on the analyzed topic. Examples:
https://pubmed.ncbi.nlm.nih.gov/11353993/
https://pubmed.ncbi.nlm.nih.gov/26499451/
Another example of an important article:
https://www.ncbi.nlm.nih.gov/pmc/articles/PMC8075307/
The introduction should contain a summary and a description of what is new for this article.
Author Response
Reviewer 2
Comment 1: The results of the statistical tests are not recorded according to scientific standards, e.g. t(24) = 3.94; p = 0.02. This applies to all tests used (chi-square, etc.).
Response: We have added t value in table 4.
Page 8, Table 4
Comment 2: The effect size for the statistical tests used has not been calculated
Response: We calculated effective size in table 3 and table 4.
Page 6-7, Table 3
Page 8, Table 4
Comment 3: Univariate and multivariate logistic regression should be performed in the analysis. The results of this analysis should be included in a summary table. Based on this, more reliable conclusions can be drawn.
Response: Thank you for this suggestion. We have performed univariate and multivariate logistic regression to analyze the poor prognostic factors associated to cerebral sinus thrombosis.
Page 6-7, Table 3
Page 8, Table 4
Comment 4: In the introduction, the authors did not include available articles on the analyzed topic.
Response: We included the papers discussed non-infectious cerebral venous thrombosis. Although infection is a risk factor for CVT, it accounts for less population when compared to non-infectious CVT, and required different treatment. Therefore, we excluded patients with infectious CVT.
Page 2, Line 75-77: Although infection is a risk factor for CVT, it accounts for less population when compared to non-infectious CVT, and required different treatment. Therefore, we excluded patients with infectious CVT.

Reviewer 3 Report
The authors made a detailed analysis of the symptoms, signs, prognosis and risk factors of rare CVT through the review of 20 years of cases, and came to the exact conclusion, which is consistent with the relevant reports and provides strong evidence support for the understanding of CVT. Overall the study is interesting, but I still have some major comments that should to be addressed.
1. The causes of CVT include trauma, infection, oral contraceptives, etc. Why did the author exclude infection factors??
2. In most available large observational studies, CVT predominantly affects women—≈75% of patients are females—and the young, with roughly 80% of patients under the age of 50. The authors' 20-year data on CVT patients showed that the male-to-female ratio and age of onset were different from those reported in most literature. Please give a reasonable explanation
3. The types of CVT include venous sinus type, cortical venous type, etc. Among them, venous sinus combined with cortical venous thrombosis is more likely to bleed, and it is also a type with poor prognosis. It is recommended to add this item to the risk factors.
Author Response
Reviewer 3
Comment 1: The causes of CVT include trauma, infection, oral contraceptives, etc. Why did the author exclude infection factors?
Response: Although infection is a risk factor for CVT, it accounts for less population when compared to non-infectious CVT, and required different treatment. Therefore, we excluded patients with infectious CVT.
Page 2, Line 75-77: Although infection is a risk factor for CVT, it accounts for less population when compared to non-infectious CVT, and required different treatment. Therefore, we excluded patients with infectious CVT.
Comment 2: In most available large observational studies, CVT predominantly affects women—≈75% of patients are females—and the young, with roughly 80% of patients under the age of 50. The authors' 20-year data on CVT patients showed that the male-to-female ratio and age of onset were different from those reported in most literature. Please give a reasonable explanation.
Response: Thank you for your comments. The sex distribution in this study is different from previous studies, suggesting unidentified selection bias, such as strategic referral from local hospitals, loss of asymptomatic patients or errors of ICD coding. Patients who have mild symptoms and spontaneous resolution may not be identified as well.
Page 9, Line 235-238: The sex distribution in this study is different from previous studies, suggesting unidentified selection bias, such as strategic referral from local hospitals, loss of asymptomatic patients or errors of ICD coding. Patients who have mild symptoms and spontaneous resolution may not be identified as well.
Comment 3: The types of CVT include venous sinus type, cortical venous type, etc. Among them, venous sinus combined with cortical venous thrombosis is more likely to bleed, and it is also a type with poor prognosis. It is recommended to add this item to the risk factors.
Response: Thank you for your comments. We characterized cortical vein thrombosis involvement in Table 3, which showed 11.7% in patients with poor clinical outcome.
Page 7, Table 3

Round 2
Reviewer 1 Report
Dear author,
The content of manuscript is significantly improved, but some concerns still present.
1. It still remains unclear, why author compares female and male (table 1 and table 2). The aim of study is to find the prognostic factors of poor (or good) outcome, that why the demographic, clinical, anatomical and laboratory features should be compare between poor and good outcome groups.
2. Figure 2 (A, B, C) shows the mRS distribution on day 7 and 30 and this distribution is compared between day 7 and day 30. Please explain what the sense of this comparison.
3. Discussion section. The author comments the risk factor of CVY in different population. The aim of this paper is not to analyze the risk factors of CVT, so this part should be removed from.
Reviewer 2 Report
The corrections have been incorporated.
The method of calculating the effect size (what are the parameters) should be described (including in the tables).
